# How to Sabbatical Successfully (and Reinvigorate Your Career)?

**DOI:** 10.3390/mps5060086

**Published:** 2022-10-25

**Authors:** Juergen K. V. Reichardt

**Affiliations:** Australian Institute of Tropical Health and Medicine, James Cook University, Smithfield, QLD 4878, Australia; juergen.reichardt@jcu.edu.au

**Keywords:** sabbatical, academic career, benefit, career planning

## Abstract

Sabbaticals should be a main feature of every academic career. This article provides some personal insights from experience along with an analysis of the benefits to the home and host institutions. Furthermore, the paper includes extensive and easy-to-follow timelines and guides for one’s own sabbatical. It is hoped that this article will expand the use of sabbaticals for everyone’s benefit in academia.

Sabbaticals are a cherished and time-honored tradition at academic institutions [1]. Unfortunately, they are also all too frequently an underutilized resource for individual career development, rejuvenation and a refreshed outlook on work, life, etc. Frankly, a sabbatical (or two) may be the difference between you keeping on doing the same old thing doing stale research and springing forward in your career and life with renewed enthusiasm in invigorated research. It appears that not much has been written about reinvigorating academic careers through sabbaticals [2,3,4]. Therefore, I am offering my own thoughts and experience here along with some practical suggestions.

Sabbaticals may also enhance institutional collaborations and research portfolios, i.e., their benefits go well beyond the individuals involved at both ends. Therefore, eligible academics worldwide should take advantage of this important perk and tool to enhance their careers and the reputation of and collaboration between the institutions involved at both ends. It is my sense that the vital importance sabbaticals can play in individual research careers as well in institutional research reputations appears generally underappreciated. Accordingly, both individual academics as well as universities should pursue and encourage sabbaticals.

Sabbaticals are generally for 1/2 year and hence restricted to a single host institution for practical reasons, including family, especially kids’ schooling. However, longer sabbaticals are possible, although often limited by the host institution to 1 year. A second institution can be considered as working from home, which became much more popular recently. I admit that I did only one sabbatical myself during my career at another institution for 1/2 year, but I learned a great deal from it, as outlined here. Specifically and firstly, that I should have gone on sabbatical sooner (and hence more often). Of course, I gained insights into a new research area and also a new friend at a different institution. Lastly, I reassessed my academic career with the benefit of distance and new surroundings.

Obviously, the major consideration is what new avenue of research benefits your program the most and who the best people in that new field are fotabler you to then work with. In my opinion, having a personal connection at the human level also helps greatly to make a sabbatical. The latter must not be preexisting, but can be developed by having a drink at a congress both parties are attending. It may also be worth considering if you are considering returning to doing benchwork yourself or rather will be analyzing data, working with students, etc.

Having lived myself in seven countries and on four continents [5], I am also a big believer in the benefits of going abroad and living in different cultures, immersing oneself in the local environment [3]. Living in new surroundings can be challenging, but in the end will be life-enriching. This also applies to your kids, who generally will adapt quickly and make new friends. Learning the basics of the local (foreign) language will make it easier and generally earn you the local’s good will just by you trying to make the effort of communicating in their language. Of course, in most labs, especially those with a number of international students and postdocs, English will be the *lingua franca*. Finally, having some basic knowledge of local customs, food and sports are most helpful and great ice breakers and conversation starters. These small efforts beforehand will make your living experience abroad that much more enriching and ease the transition into a new world.

Therefore, I thought I would lay out here my suggestions and proposed time lines to organize a successful sabbatical for academics interested in pursuing one in Table 1, Table 2, Table 3, Table 4 and Table 5 at an academic host institution. Sabbaticals in industry [2] or perhaps even at home are an option, especially in the post-pandemic world. Obviously, a sabbatical does not materialize overnight, but rather requires some significant planning and preparation. However, with all pieces in place correctly, a sabbatical can be a roaring success and the jolt your career may need to keep moving forward successfully for all parties involved. I will focus on research due to my own experience, but note that others have published recently on the experiences of clinicians [6], nurses [7] and more [8,9,10]. I note that sabbaticals are not easily accessible to all [10]. This situation deserves attention, especially in light of the increasing casualization of the academic work force [11].

In summary, consider a sabbatical early and then repeat as you move through your career. Enjoy your sabbaticals and return refreshed to your home institution and work. Sabbaticals should be a regular feature in academic careers and a valuable tool throughout career progression. Others have made similar points [2]. Universities should encourage their staff to engage in sabbaticals regularly, benefitting both the academics involved and the institutions at both ends.

## Figures and Tables

**Table 1 mps-05-00086-t001:** 2–3 years before your Planned sabbatical.

Identify a new area of research that complements and enhances your current research portfolio
Talk to suitable colleagues worldwide to host you for your sabbatical
Begin thinking jointly of a suitable research project (and logistics)
Engage with your chair on your planned sabbatical for support departmentally, faculty- and university-wide

**Table 2 mps-05-00086-t002:** 1–2 years before your sabbatical.

Complete the necessary paperwork for your sabbatical at your institution
Consider and plan for children’s and partner’s needs, e.g., schooling, work, etc.
Begin discussing the distribution of your teaching, administrative, etc. activities during your sabbatical (remembering that one hand washes another, i.e., you may be called upon for someone else’s sabbatical)
Begin learning the basics of the local language and culture (food, customs, sport, etc.)

**Table 3 mps-05-00086-t003:** In the year before your sabbatical.

Organize substitute teaching, admin, etc., as warranted
Set up a trusted and competent representative for your lab in your absence (practice makes perfect, i.e., let them meet with lab members whilst you are still around and can fix any issues that might pop up)
Firm up the research project for your sabbatical with your host
Complete any paperwork required by your host institution
Have a valid passport if applicable
Get a visa if necessary
Rent out your place if warranted
Obtain suitable housing at the place of your sabbatical
Organize schooling, etc., for your kids if they are accompanying you

**Table 4 mps-05-00086-t004:** During your sabbatical.

Enjoy the experience, including life in a new cultural environment (if going abroad)
Learn lots of new stuff (in the lab, life, etc.)
Soak up the new atmosphere at your host institution (thinking of what is better, the same and perhaps appreciate what is better at your home university)
Stay on top of your lab and ongoing research at your home institution
Keep in touch with your students, postdocs, staff, etc. you mentor through regular (e.g., zoom) meetings

**Table 5 mps-05-00086-t005:** After your sabbatical.

Resume your normal teaching and administrative duties, rejuvenated and with a fresh perspective
Keep in touch with your sabbatical colleagues, firm up and expand collaboration
Write papers, grants proposals, etc., stemming from your sabbatical
Build up your new research area resulting from your sabbatical

## Data Availability

Not applicable.

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
