# Peer review of "How to Sabbatical Successfully (and Reinvigorate Your Career)?"

_mps, 2022, doi:10.3390/mps5060086_

Round 1
Reviewer 1 Report
I don't know this journal, but I do know personal opinion is not really that relevant in a research journal.
Consider increasing the research into sabbatical benefits, sabbaticals themselves...
Author Response
Please see attached letter.

Reviewer 2 Report
I think this opinion piece does contain some useful practical advice about preparing and doing sabbaticals, but I am concerned about the implicit assumptions about equity of access, and a lack of drawing on past research about sabbaticals. The author admits they have only had one sabbatical – for a piece written about the value of sabbaticals and practical advice, I would have thought more experience would be a prerequisite, or at least talking with colleagues, and undertaking a literature search, to get a wider range of experience represented.
Really I wanted to see acknowledgement that sabbaticals can take different forms e.g., stay at home versus going abroad, and, for the latter, recognition that the academic can visit more than one institution. If just focusing on going abroad (and only for expanding research) and to only one institution, this focus should be made clear earlier on (and amidst recognition of other forms). Or, even better a comparison of the pros and cons of different forms of sabbaticals, including considerations of equity and diversity might mean some forms are more accessible than others.
It is necessary to delve into the literature (for which there is a modest body) to probe benefits and challenges in an evidence-based way rather than relying solely on one’s own, single experience. This would help provide a more compelling set of guidelines about how to prepare and undertake sabbaticals, and hopefully make your recommendations more widely accessible to all.
Author Response
- Please see the attached letter.

Reviewer 3 Report
There is a rather large, and growing body of literature and research on the sabbatical that includes processes for narrowing topics and being strategic in setting one up. This research would enable the article to move from being a personal reflection to something that might carry more weight in the planning process.
Author Response
- Please see the attached letter.

Round 2
Reviewer 2 Report
Although some minor changes have been made, I am still unable to support this for publication. The major limitation is a lack of in-depth consideration of relevant research and the fact the opinion is based on only one sabbatical. Issues of equity of access are still not considered. There is simply not enough evidence-based content to warrant publication.
Author Response
- I have added in additional references as recommended noting that the paper is really a guide and hence suitable for a journal such as Methods and Protocols. Literature should, of course, always be referenced appropriately but this article is not a review. Rather it is a guide as just noted.
- Furthermore, I have added considerations of equity.
- Finally, I have made minor improvements to the write-up as suggested.
Reviewer 3 Report
There is certainly a great deal more literature on the sabbatical that could be included.
Author Response
- I have included further references as suggested.
- Spell checking was also performed as recommended.
Round 3
Reviewer 2 Report
While it is good to see some recognition of wider literature and access issues, I am sorry but I still do not see this as meeting publication standard. Even an opinion piece should be firmly grounded in relevant literature, especially when relying on only one experience of a sabbatical. When the reviewers had suggested major changes, I expect to see a much deeper engagement with the feedback, rather than the superficial adding in of more references.
Author Response
Please find attached the previously revised manuscript for MPs. I appreciate the dissenting referee’s time and comments but emphatically disagree as per the below:
Reviewer 2:
- I appreciate this referee’s time and the elaboration on the foundations of scientific literature but disagree specifically as follows: the journal at hand is called Methods and Protocols and not Reviews of….. Thus, the emphasis is on practical guidelines and not on an exhaustive literature review. I believe the current manuscript fulfills the remit of the journal.
- As a sign of good will, I am, however, happy to consider some citations this reviewer may wish to suggest for inclusion. Obviously, as the author, I will be the judge citing any of the suggested references.
- Finally, I am appalled by the tone of this referee and suggest a professional and collegial exchange.
I am happy to answer questions you may have and look forward to hearing from you in due time.
With kind regards for your further consideration.